# Transport of Chromium(VI) across a Supported Liquid Membrane Containing Cyanex 921 or Cyanex 923 Dissolved in Solvesso 100 as Carrier Phase: Estimation of Diffusional Parameters

**DOI:** 10.3390/membranes13020177

**Published:** 2023-02-01

**Authors:** Francisco J. Alguacil, Jose I. Robla

**Affiliations:** Centro Nacional de Investigaciones Metalurgicas (CENIM-CSIC), Avda. Gregorio del Amo 8, 28040 Madrid, Spain

**Keywords:** facilitated transport, Cyanex 921, Cyanex 923, chromium(VI), hydrazine sulphate

## Abstract

An investigation of chromium(VI) transport across a supported liquid membrane containing the phosphine oxides Cyanex 921 and Cyanex 923 dissolved in Solvesso 100 as carrier phases was carried out in batch operation mode. Chromium(VI) transport was investigated as a function of different variables: hydrodynamic conditions in the feed (1000–1600 min^−1^) and stripping (600–1500 min^−1^) phases, HCl (0.25–2 M) and indium (0.01–0.1 g/L) concentrations in the feed phase, and carrier (0.01 M–0.75 M) concentration in the membrane phase. Indium was recovered in the stripping phase using hydrazine sulphate solutions, and, at the same time, chromium(VI) was reduced to the less harmful Cr(III) oxidation state. Models describing the transport mechanism comprising a diffusion process through the feed aqueous diffusion layer, fast interfacial chemical reaction, and diffusion of the respective chromium(VI)–phosphine oxide complexes across the membrane were developed. The equations describing the rate of transport correlate the membrane permeability coefficient with diffusion and equilibrium parameters, as well as the chemical compositions of the respective metal–carrier phases. The models were used to calculate diffusional parameters for each metal–carrier system, and the minimum thickness of the feed boundary layer was calculated as 1 × 10^−3^ cm and 6.3 × 10^−4^ cm for the Cr(VI)-Cyanex 921 and Cr(VI)-Cyanex 923 systems, respectively.

## 1. Introduction

Chromium, especially chromium(VI), is a very hazardous element to humans, due to the various diseases, including cancer, that its ingestion can cause [1,2,3,4,5]. However, this element is widely used in a range of industries, in electroplating, pigment production, leather tanning, metal finishing, electrical and electronic equipment, catalysis, etc.; moreover, galvanizing industries also use chromium(VI) to passivate zinc, increasing corrosion resistance and at the same time altering the color and appearance of steel. The presence of the element in liquid effluents generated from the abovementioned industries has adverse effects on humans and the environment, such that its removal from the effluents containing it is of the utmost importance. It is worth noting here that limits on the presence of chromium(VI) (also total chromium) in water or wastewater vary from one country to another. The US EPA fixed the limit of the presence of total chromium in drinking water at 0.1 mg/L [6], whereas in Europe legislation is more flexible regarding the presence of total chromium and chromium(VI) in wastewater [7], e.g., Spain has set limits of 5 mg/L and 0.3 mg/L, Croatia limits of 1–4 mg/L and 0.1 mg/L, Italy 2–4 mg/L and 0.1 mg/L and Germany 0.1–0.5 mg/L and 0.05–0.5 mg/L, respectively. From the above, it is clear that the recovery of chromium(VI) from different materials and generated wastes can be included in the concept of urban mining.

Several separation technologies are used or have been proposed to eliminate chromium(VI) from various effluents [8]. These technologies include solvent extraction, membranes, and adsorption; photocatalysis and nanotechnologies also have promising futures in this field.

In the case of solvent extraction, the use of TOPO (solvation extractant), Cyanex 272 (alkylphosphinic acid extractant), and mixtures of both reagents have been described for the removal of this element from solutions [9], while the extraction and separation of vanadium(V) and chromium(VI) have also been investigated using amides of mixtures of amides and Cyanex 272 [10,11,12]. 

Graphene membranes [13] are used to remove chromium(VI) from solutions of pH 2. In the process, chromium(VI) is mainly reduced to the chromium(III) oxidation state. Polyvinylamine-grafted polypropylene membranes are also used to remove chromium(VI) from solutions with acidic pH values [14]; in such cases, chromium(VI) is removed from the membrane by means of NaOH/NaCl aqueous solutions.

Metallic nanoparticles are also used to remove chromium(VI) from solutions via reduction of the (VI) oxidation state to the corresponding (III) state [15].

Judging from the number of recently published works, adsorption seems to be the technology most widely used to investigate the removal of chromium(VI) from solutions. The investigated adsorbents have included: a carbon dot–chitin nanocrystal hybrid [16], polystyrene microplastics [17], CdO nanoparticles formed into grapheme and grapheme oxide nanosheets [18], fungal biomass [19], biochar nanocomposites based on CoFeO_4_ [20], activated carbon from oil shale [21], aerogel composites from chitosan and pineapple-leaf-based cellulose [22], an eggshell–polypyrrole composite [23], and activated carbon from guava seeds [24]. Besides these *conventional* adsorption methodologies, there is also great interest in investigating the removal of toxic chromium(VI) from solutions via reduction to the less harmful chromium(III) oxidation state. Among the adsorbents used for this purpose, published investigations have used: nanozerovalent iron in a nonaqueous medium [25], nanozerovalent iron particles immobilized on Mbenes [26], nanozerovalent iron-rgGO hybrids [27], nanoscale FeS assembled onto MXenes [28], Cu_2_ZnSnS_4_ (containing Ag or Au) nanocrystals [29], and sulphonated polymers [30].

Included among the abovementioned separation technologies, liquid membrane processes have been of increasing interest. Liquid membrane processing is rate-governed rather than having equilibria between phases; this means that metal can be transported along a chemical gradient and also against a gradient. There are three main configurations of liquid membranes: bulk liquid membranes, emulsion liquid membranes, and supported liquid membranes; and whereas the latter use a (hydrophilic) solid support to retain the carrier phase and separate the feed and stripping phases, the other two do not use such supports [31]. From an engineering and practical point of view, supported liquid membranes are of particular interest due to their stability (if correctly utilized) and simplicity, as the extraction and stripping stages are combined in a single operation [32]. These liquid membrane technologies have also recently been used to investigate the removal of chromium(VI) from aqueous media. A liquid emulsion membrane system, using Aliquat 336 (quaternary ammonium salt) dissolved in a diesel diluent and Span 80 as an emulsifier, was used to investigate the removal of this toxic metal from wastewater [33]; the removal process was aided by applying hydrodynamic cavitation. Using a liquid surfactant membrane technique [34], the removal of chromium(VI) from both a synthetic solution and real galvanizing industry wastewater was investigated; in this study, the tertiary amine Alamine 336 was used to extract the toxic metal. A flat-sheet-supported liquid membrane containing DEHPA (organophosphoric acid derivative) as a carrier was used to investigate the separation of chromium(VI), copper(II), and zinc(II) [35]. Being an acidic extractant, it is obvious that chromium(VI) (forming anionic species in the solution) remained in the aqueous feed solution, whereas copper(II) and zinc(II) (both present as cations in the feed solution) were transported by this carrier. Under the same flat-sheet operational mode but in a dispersion-supported liquid membrane mode [36], the transport of chromium(VI) was investigated using a tertiary amine (N235) as a carrier. Under the best operational conditions, 94% of the metal, from an original feed of 0.01 g/L, could be transported. 

In order to improve general knowledge of the use of supported liquid membranes for the removal of this toxic metal from aqueous solutions, the present work used a flat-sheet-supported liquid membrane impregnated with Cyanex 921^−^ or Cyanex 923 dissolved in Solvesso 100 to investigate the active transport of chromium(VI) from acidic solutions. The influence of various experimental variables on the transport of chromium(VI) by these carrier phases is described and, from the experimental data, diffusional parameters were estimated for both systems: Cr(VI)-Cyanex 921-Solvesso 100^−^ and Cr(VI)-Cyanex 923-Solvesso 100. 

## 2. Materials and Methods

### 2.1. Materials

The phosphine oxides Cyanex 921 and Cyanex 923 (Solvay, France) were used without further purification; they have the following compositions [37]: Cyanex 921 has tri-n-octylphosphine oxide as an active group, whereas Cyanex 923 is composed of four tryalkyl phosphine oxides. Besides their composition, the main difference between these two reagents is that Cyanex 921 is a solid whereas Cyanex 923 is a liquid, which removes any potential solubility problem when it is in the presence of organic diluents; in fact, Cyanex 923 has the potential to be used undiluted. 

A quantity of 1 g/L chromium(VI) stock solution was prepared by dissolving potassium dichromate (Fluka, Switzerland) in distilled water, and working solutions were prepared from dilutions of the above. All the chemicals used in the experimentation, except the phosphine oxides and Solvesso 100 (aromatic diluent), were of AR grade. The solid support used in the present work was Millipore Durapore GVHP4700 (polyvinylidene fluoride) (75% porosity, 1.67 tortuosity, 12.5 × 10^−3^ cm thickness).

### 2.2. Methods

#### 2.2.1. Transport

Chromium(VI) transport experiments were carried out in a two-compartment cell, consisting of a feed phase (200 cm^3^) separated from a stripping phase (200 cm^3^) by a membrane support. The effective membrane area for the transport experiments was 11.3 cm^2^. Both feed and stripping phases were mechanically stirred by four-blade glass impellers (2.5 cm diameter) at 20 °C to avoid concentration polarization conditions at the support interfaces and in the bulks of both phases [38]. 

The supported liquid membrane was prepared by impregnation of the solid support with the corresponding organic phase by immersion for 24 h and was then left to drip for 20 seconds before it was placed in the membrane cell. Previous tests demonstrated that an extended immersion time did not influence chromium transport.

The overall mass transfer coefficient (K_o_) was determined by monitoring the metal concentration in the feed phase at elapsed times by atomic absorption spectrometry (Perkin Elmer 1100B spectrophotometer, United Kingdom), using the following equation:(1)$$\ln\frac{{\left[\mathrm{Cr}\right]}_{f,t}}{{\left[\mathrm{Cr}\right]}_{f,0}}=-\frac{A}{V}K_{o}t$$
where [Cr]_f,0_ and [Cr]_f,t_ are the metal concentrations in the feed phase at time zero and at an elapsed time, respectively; A is the membrane area; V is the volume of the feed phase; and t represents the elapsed time. The percentage of chromium recovered from the membrane phase to the receiving phase was calculated using the following equation:(2)$$\%R=\frac{{\left[\mathrm{Cr}\right]}_{r,t}}{{\left[\mathrm{Cr}\right]}_{f,0}-{\left[\mathrm{Cr}\right]}_{f,t}}\times 100$$
where [Cr]_r,t_ represents the indium concentration in the receiving phase at an elapsed time.

#### 2.2.2. Distribution Coefficient of Chromium(VI)

Determinations of the distribution coefficients were carried out using equal volumes (20 mL) of aqueous phases containing 0.1 g/L and 1 M HCl and organic phases containing 0.01–0.1 M extractant (Cyanex 921 or Cyanex 923) dissolved in Solvesso 100. The various phases were mechanically shaken at 20 °C and 15 min (a sufficient length of time for equilibria to be reached). After phase separation, the chromium was analyzed (in a way similar to that described above) in the equilibrated aqueous solution, and the content in the organic phase was calculated by mass balance. The distribution coefficient (D_ext_) of Cr(VI) for extraction, defined as the ratio of the total metal concentration in the organic phase [Cr]_org_ and in the aqueous phase [Cr]_aq_, was calculated using the following formula:(3)$$D_{\mathrm{ext}}=\frac{{\left[\mathrm{Cr}\right]}_{\mathrm{org}}}{{\left[\mathrm{Cr}\right]}_{\mathrm{aq}}}$$

## 3. Results and Discussion

The transport of chromium(VI) across the support containing the phosphine oxides in Solvesso 100 can be described by Fick’s first diffusion law, from the diffusion layer at the feed-phase side to the membrane phase and to the stripping phase. However, this last contribution is often negligible compared with that at the feed-phase side since the distribution coefficient of the metal between the membrane and the stripping phases is lower than the value between the feed and the membrane phases. Figure 1 shows the concentration profiles of the chromium and the phosphine oxides dissolved in Solvesso 100 during the metal-transport process. Thus, in the transport of chromium(VI), the driving force is the difference in acidity between the feed and stripping phases.

### 3.1. Transport of Chromium(VI) Using Cyanex921-Solvesso 100 Solutions as Carrier Phases

#### 3.1.1. Influence of Stirring Speed (Feed Phase)

To reach the effective permeation of chromium(VI) in a supported liquid membrane system, it is convenient to investigate the influence of the stirring speed on the overall mass transfer coefficient. In the present work, stirring of the feed phase varied in the 1000–1500 min^−1^ range; whereas the feed phase contained 0.01 g/L Cr(VI) in 1 M HCl, the membrane phase consisted of 0.5 M Cyanex 921 in Solvesso 100 solution immobilized on a GVHP4700 support, and the stripping phase contained a solution of 10 g/L hydrazine sulphate. The experimental results (Figure 2) indicated that within this range of stirring speed the overall mass transfer coefficient was constant (K_o_ = 1.1 × 10^−2^ cm/s). In a transport process across a supported liquid membrane, diffusional resistances are of two types: (i) resistance due to the feed-phase boundary layer and (ii) resistance associated with the membrane support. It is not rare to find that the magnitude of the first competes with the value of the support resistance [39]. These experimental results showed that, in the 1000–1500 min^−1^ range, the aqueous boundary layer reached a minimum and that the aqueous resistance to mass transfer was minimized; thus, the diffusion contributions of the aqueous species to the mass-transfer phenomena were considered constant [40].

#### 3.1.2. Influence of Stirring Speed (Stripping Phase)

Using the same experimental conditions as described in the preceding subsection and maintaining a stirring speed of 1400 min^−1^ in the feed phase, it was concluded that the variation in the stirring speed in the stripping phase, within the same 1000–1500 min^−1^ range, had a negligible influence on the value of the overall mass transfer coefficient (see above). In the case of the stripping phase, and if the stirrer in the half-cell was close to the membrane support, the thickness of the boundary layer was considered to be minimized and the resistance in this side could be neglected [41]. Thus, stirring speeds of 1400 min^−1^ in the feed and stripping phases were used throughout all the experimental work. It is worth noting here that after 2 hours of elapsed time the percentage of chromium recovered in the stripping phase reached 85%, indicating the usefulness of hydrazine sulphate as a strippant for the system.

#### 3.1.3. Influence of Stripping Phase Composition

The effect of variation in hydrazine sulphate concentration in the stripping phase on chromium(VI) transport was also evaluated. The feed phase was composed of 0.01 g/L Cr(VI) in 1 M HCl, whereas the membrane phase contained 0.25 M Cyanex 921 in Solvesso 100 immobilized on a GVHP4700 support. The results indicated that there was a slight increase in metal transport when the concentration of hydrazine sulphate in the stripping phase was increased from 2.5 g/L (K_o_ = 8.7 × 10^−3^ cm/s) to 5 g/L (K_0_ = 1.0 × 10^−2^ cm/s), whereas the variation in K_o_ when the strippant concentration was raised from 5 g/L to 10 g/L was negligible. In this stripping phase, the reduction of Cr(VI) to Cr(III) was due to the following reaction:(4)$$4H_{2}\mathrm{Cr}O_{4}\cdot nL_{m}+3N_{2}H_{4}\cdot H_{2}SO_{4_{\mathrm{st}}}\to 4Cr_{\mathrm{st}}^{3+}+3SO_{4_{\mathrm{st}}}^{2-}+6OH_{\mathrm{st}}^{-}+3N_{2}+10H_{2}O_{\mathrm{st}}+4nL_{m}$$
where the subscripts m and st represent the membrane and stripping phases.

#### 3.1.4. Influence of HCl Concentration (Feed Phase)

The effect of the variation (0.25–2 M) in HCl concentration, in the feed phase, on chromium(VI) transport was also investigated using membrane phases of 0.5 M Cyanex 921 in Solvesso 100 immobilized on a GVHP4700 support and stripping phases of 5 g/L hydrazine sulphate. The results of these experiments are summarized in Table 1.

It can be seen that the increase in HCl concentration (from 0.25 M to 0.75 M) in the feed phase improved the transport of this metal; however, in the 0.75–2 M range, no further improvement in chromium(VI) transport was observed.

#### 3.1.5. Influence of Extractant Concentration (Membrane Phase)

Investigations of the influence of variation in Cyanex 921 concentration on metal transport in the membrane phase were also performed. The feed phase was composed of 0.01 g/L Cr(VI) in 1 M HCl, and 5 g/L hydrazine sulphate solutions were used as strippants. The results are summarized in Table 2.

It can be seen that there was a continuous increase in chromium transport when the extractant concentration increased in the 0.01–0.13 M range, transport being constant when higher extractant concentrations were used.

These results can be explained by the fact that in the 0.01–0.13 M extractant concentration range, membrane diffusion governed metal transport and that in the 0.13–0.5 M range aqueous diffusion was responsible for the transport of chromium(VI) from the feed to the membrane phase. In these conditions (0.13–0.5 M), a limiting overall mass transfer coefficient is defined as [42]:(5)$$K_{O}^{\lim}=\frac{D_{f}}{d_{f}}$$
and, assuming an average value of 10^−5^ cm^2^/s for the metal aqueous diffusion coefficient (D_f_) and a value of 1.0 × 10^−2^ cm/s for the limiting overall mass transfer coefficient, a value of 1.0 × 10^−3^ cm can be estimated for d_f_, this value being the minimum thickness of the feed-phase diffusion layer.

#### 3.1.6. Influence of Initial Chromium(VI) Concentration in the Feed Phase

Table 3 shows the experimental conditions and overall mass transfer coefficient values at different initial chromium(VI) concentrations in the feed phase.

The results indicated that chromium(VI) transport decreased with the increase in the initial metal concentration in the feed phase; thus, at lower metal concentrations the transport process is controlled by the diffusion of metal species, and the flux may increase with the increase in metal concentration in this phase.

### 3.2. Transport of Chromium(VI) Using Cyanex923-Solvesso 100 Solutions as Carrier Phases

#### 3.2.1. Influence of Stirring Speed (Feed Phase)

To investigate the influence of this variable on chromium(VI) permeation across the supported liquid membrane containing Cyanex 923 in Solvesso 100, feed solutions of 0.01 g/L Cr(VI) in 1 M HCl, membrane phases of 0.5 M Cyanex 923 in the organic diluent immobilized on a GVHP4700 support, and stripping phases of 5 g/L hydrazine sulphate were used. The stirring speed (feed phase) was varied between 1000 and 1600 min^−1^, whereas the stirring speed in the stripping phase was maintained at 1400 min^−1^. The results for these experiments (Figure 3) showed that in the 1200–1600 min^−1^ range the overall mass transfer coefficient values reached a maximum of 1.6 × 10^−2^ cm/s.

#### 3.2.2. Influence of Stirring Speed (Stripping Phase)

With the stirring speed of the feed phase maintained at 1400 min^−1^ and using the same experimental conditions as described in Section 3.2.1, it was shown that the variation in the stirring speed (600–1500 min^−1^) applied on the strip phase had a negligible effect on chromium(VI) transport above 1000 min^−1^ (K_o_ = 1.6 × 10^−2^ cm/s). The percentage of chromium recovered in the strip phase, as chromium(III), reached 90%, indicating that the use of this strippant is convenient in this system because (i) it allows a good recovery of chromium in the strip phase and (ii) chromium is reduced to the less harmful Cr(III) oxidation state. In all the subsequent experimentation, stirring speeds of 1400 min^−1^ were used in both the feed and stripping phases.

#### 3.2.3. Influence of Stripping Phase Composition

Maintaining all the experimental conditions as in the previous sections but using different concentrations of the strippant in the corresponding phase, the results revealed that there was an increase in chromium(VI) transport from solutions containing 1 g/L to 5 g/L of the strippant and that, within this concentration range, metal transport remained unaltered, with overall mass transfer coefficient values being close to 1.6 × 10^−2^ cm/s. As previously described, Equation (4) explains the metal-reduction process.

#### 3.2.4. Variation in HCl Concentration in the Feed Phase

In this set of experiments, feed phases contained 0.01 g/L Cr(VI) in various (0.25 to 2 M) HCl media, whereas the membrane phase contained 0.5 M Cyanex 923 in Solvesso 100 immobilized on a GVHP4700 support. A 5 g/L hydrazine sulphate solution was used as the stripping phase. The results derived from this set of experiments showed that in the 0.25–1 M HCl range chromium(VI) transport was unaltered (K_o_= 1.6 × 10^−2^ cm/s), this value decreasing to 7.5 × 10^−3^ cm/s when the hydrochloric acid concentration in the feed phase was 2 M. The existence of CrO_3_Cl^−^ in HCl solutions has been described in the literature [43], and probably the observed decrease in metal transport was due to the lower extractability of this compound by Cyanex 923; however, in the case of Cyanex 921 (see Section 3.1.4) this effect has not been noted.

#### 3.2.5. Influence of Cyanex 923 Concentration on Chromium(VI) Transport

In the investigation of the influence of this variable on chromium(VI) transport, feed phases of 0.01 g/L Cr(VI) in 1 M HCl and stripping phases of 5 g/L hydrazine sulphate were used. The membrane phase contained various (0.02 to 0.75 M) Cyanex 923 concentrations in Solvesso 100. The results (Figure 4) showed that there was a continuous increase in chromium(VI) transport as the carrier concentration in the membrane phase increased; however, in the 0.13–0.75 M Cyanex 923 concentration range, metal transport reached a maximum.

These results also show that in the 0.02–0.13 M range, chromium(VI) transport was governed by membrane diffusion and that from 0.13 M to 0.75 M the influence of membrane diffusion was negligible and that the transport was controlled by aqueous diffusion. In the 0.13–0.75 M Cyanex 923 concentration range, the overall mass transfer coefficient reached a maximum value (K_o_ = 1.6 × 10^−2^ cm/s), and, using the same formula (Equation (5)), the minimum thickness of the aqueous boundary layer was calculated as 6.3 × 10^−4^ cm.

#### 3.2.6. Effect of Varying the Initial Chromium(VI) Concentration in the Feed Phase on Metal Transport

Membrane phases of 0.5 M Cyanex 923 in Solvesso 100 immobilized on GVHP4700 supports and stripping phases of 5 g/L hydrazine sulphate were used to investigate the influence that variation in chromium(VI) concentration in the feed phase has on metal transport. In this set of experiments, the feed phases contained various (0.01 to 0.08 g/L) chromium(VI) concentrations in 1 M HCl medium. The results derived from the above experiments are shown in Figure 5, plotting the overall mass transfer coefficient values versus initial chromium(VI) concentrations in the feed phases.

This figure shows that there was a continuous decrease in the overall mass transfer coefficient values as the initial metal concentrations in the feed phases increased. Thus, similar to what was described in Section 3.1.6, the diffusion of metal species governed metal transport, especially with low chromium(VI) concentrations in the feed solutions.

### 3.3. Estimation of Diffusional Parameters in the Cr(VI)-Cyanex 921-Solvesso 100 and Cr(VI)-Cyanex 923-Solvesso 100 Transport Systems

In the first instance, the extraction (transport) of chromium(VI) by these two phosphine oxides can be represented by the following equilibrium expression (see also Figure 1):(6)$$\mathrm{HCr}O_{4_{\mathrm{aq}}}^{-}+H_{\mathrm{aq}}^{+}+nL_{\mathrm{org}}\Leftrightarrow H_{2}\mathrm{Cr}O_{4}\cdot nL_{\mathrm{org}}$$
where L represents the active groups of Cyanex 921 or Cyanex 923 extractants and the subscripts aq and org represent the aqueous (feed) and organic (membrane) phases, respectively.

The extraction constant of the above equilibrium can be expressed as:(7)$$K_{\mathrm{ext}}=\frac{{\left[H_{2}\mathrm{Cr}O_{4}\cdot \mathrm{nL}\right]}_{\mathrm{org}}}{{\left[\mathrm{HCr}O_{4}^{-}\right]}_{\mathrm{aq}}{\left[H^{+}\right]}_{\mathrm{aq}}{\left[L\right]}_{\mathrm{org}}^{n}}$$

Taking into account the definition of the distribution coefficient (Equation (3)) and substituting it in Equation (7), rearranging and taking logarithms, the following expression relates both the extraction constant and the distribution coefficient:(8)$$\log D_{\mathrm{ext}}=\log K_{\mathrm{ext}}+\log{\left[H^{+}\right]}_{\mathrm{aq}}+\mathrm{nlog}{\left[L\right]}_{\mathrm{org}}$$

At 1 M HCl, the final expression is:(9)$$\log D_{\mathrm{ext}}=\log K_{\mathrm{ext}}+\mathrm{nlog}{\left[L\right]}_{\mathrm{org}}$$

Experimental results derived from the extraction experiments at various L concentrations allowed us to estimate the value of log K_ext_ and the coefficient n for both extraction systems (Table 4) by plotting log D_ext_ versus log [L]_org_.

Moreover, the treatment of the experimental results was refined numerically by a tailored computer program which minimized the U-values, defined as:(10)$$U=\Sigma {\left(\log D_{\mathrm{ext},\exp}-\log D_{\mathrm{ext},\mathrm{calc}}\right)}^{2}$$
where D_ext,exp_ and D_ext,calc_ are the values of the experimental distribution coefficients calculated by the program. The results of these numerical calculations are also given in Table 4. It can be seen that, in all the cases, the stoichiometric factor n is given as 2 or near 2; thus, the extracted species presented stoichiometries of H_2_CrO_4_·2Cyanex 921 and H_2_CrO_4_·2Cyanex 923. The value of this stoichiometric factor is similar to those described in the literature for systems involving chromium(VI) and phosphine oxides [9,43]. It is worth noting here that the extraction of metal species with phosphine oxides is related to a solvation reaction, in which the phosphine oxide donates an electron pair (via the oxygen atom of the extractant molecule) to the metal, substituting the water hydration molecules of the inorganic moiety.

Given the same assumptions as those described in the literature [44], an expression such as:(11)$$K_{o}=\frac{K_{\mathrm{ext}}{\left[L\right]}_{m}^{2}}{\Delta_{m}+\Delta_{f}\left(K_{\mathrm{ext}}{\left[H^{+}\right]}_{f}{\left[L\right]}_{m}^{2}\right)}$$
combines the equilibrium and diffusion parameters involved in the transport of chromium(VI) from 1 M HCl solutions across a membrane support containing Cyanex 921 or Cyanex 923 dissolved in Solvesso 100. In this Equation (11), the subscripts m and f represent the membrane (organic) phase and the feed (aqueous) phase, respectively.

Rearrangement of the above equation gives:(12)$$\frac{1}{K_{o}}=\Delta_{f}+\Delta_{m}\frac{1}{K_{\mathrm{ext}}{\left[H^{+}\right]}_{f}{\left[L\right]}_{m}^{2}}=\Delta_{f}+\Delta_{m}\frac{1}{A}$$
where Δ_f_ and Δ_m_ are the mass transfer resistances in the feed and membrane phases, respectively. Using various extractant concentrations and plotting 1/K_o_ versus 1/A may result in straight lines with ordinates to calculate Δ_f_ and the slope of Δ_m_. These values and other diffusional parameters calculated (see below) using Cyanex 921 or Cyanex 923 in the facilitated transport of chromium(VI) are summarized in Table 5.

The membrane diffusion coefficient (D_m_) is related to the membrane thickness (d_m_) and the mass transfer resistance due to the membrane phase by:(13)$$\Delta_{m}=\frac{d_{m}}{D_{m}}$$

The diffusion coefficient of chromium(VI)-Cyanex 921 or chromium(VI)-Cyanex 923 species in the bulk organic phase can be calculated by [45]:(14)$$D_{b,m}=D_{m}\frac{\tau^{2}}{\varepsilon }$$

If it is considered that the carrier concentrations in the membrane are constant, the apparent diffusion coefficient for chromium(VI) can be calculated as:(15)$$D_{m}^{a}=\frac{J_{\mathrm{Cr}}d_{m}}{{\left[L\right]}_{m}^{2}}$$
where J_Cr_ is the metal flux (J_Cr_ = K_o_[Cr]_f,0_).

Under some circumstances, i.e., high metal concentrations in the feed phase, and due to the rate-determining step for the transport process, the metal flux can reach a limit; this limiting flux (J_lim)_) can be calculated using the following expression [46]:(16)$$J_{\lim}=\frac{D_{m}{\left[\mathrm{Cr}(\mathrm{VI})-L\right]}_{m}}{d_{m}n}$$
where n is the stoichiometric factor of the extraction reaction and [Cr(VI)-L]_m_ is the concentration of the corresponding Cr(VI)-phosphine oxide compound. Under these limiting conditions, this concentration is equivalent to the total concentration of the extractant in the organic phase.

Table 5 summarizes the values of the diffusional parameters calculated for the Cr(VI)-Cyanex 921-Solvesso 100 and Cr(VI)-Cyanex 923-Solvesso 100 transport systems immobilized on GVHP400 supports.

As shown in Table 5, the diffusion coefficient in the bulk organic phase presented a greater value than the membrane diffusion coefficient, this being attributable to the diffusional resistance caused by the support thickness separating the feed and stripping phases.

## 4. Conclusions

This work demonstrated that the phosphine oxides Cyanex 921 and Cyanex 923 can be used as carrier phases for chromium(VI) membrane transport from hydrochloric acid solutions.

The transport of chromium(VI) is influenced by a series of variables, such as the stirring speed applied on the feed phase and the concentrations of metals and their respective carriers. In the case of Cyanex 923, the variation in HCl concentration above 1 M in the feed phase also influenced the transport of this toxic metal. Using both phosphine oxides, with carrier concentrations of 0.13 M in Solvesso 100, a maximum level of chromium(VI) transport was obtained. Under these conditions, the transport process was controlled by diffusion in the feed phase; at carrier concentrations lower than this 0.13 M concentration, membrane diffusion controlled the overall metal transport.

In addition, with the two phosphine oxides, the extracted (transported) species had the same stoichiometry H_2_CrO_4_·2L (L representing the active group of Cyanex 921 or Cyanex 923).

Experimental data indicated that hydrazine sulphate solutions can be used as effective stripping phases in both transport systems. The use of this chemical also promotes the reduction of chromium(VI) to the less hazardous chromium(III) oxidation state.

Various diffusional parameters involved in the metal permeation were calculated for both Cr(VI)-Cyanex 921 and Cr(VI)-Cyanex 923 transport^−^systems.

## Figures and Tables

**Figure 1 membranes-13-00177-f001:**
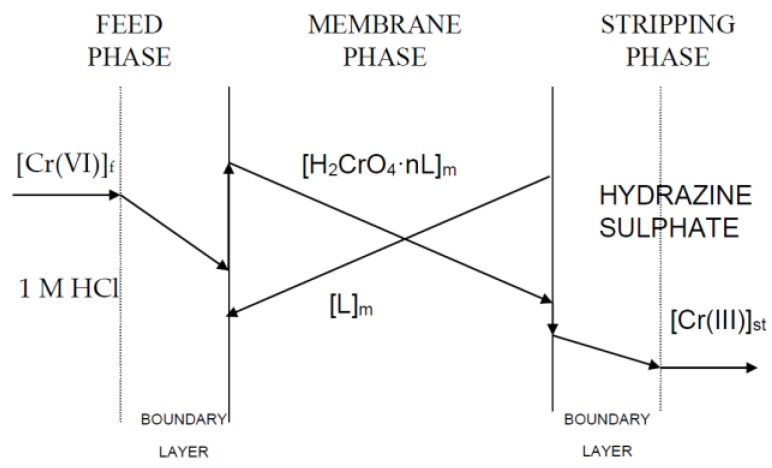
Concentration profiles of the chromium and phosphine oxide (L) species in the metal-transport system. Feed phase: f. Membrane phase: m. Stripping phase: st. Stoichiometric coefficient: n (determined in Section 3.3).

**Figure 2 membranes-13-00177-f002:**
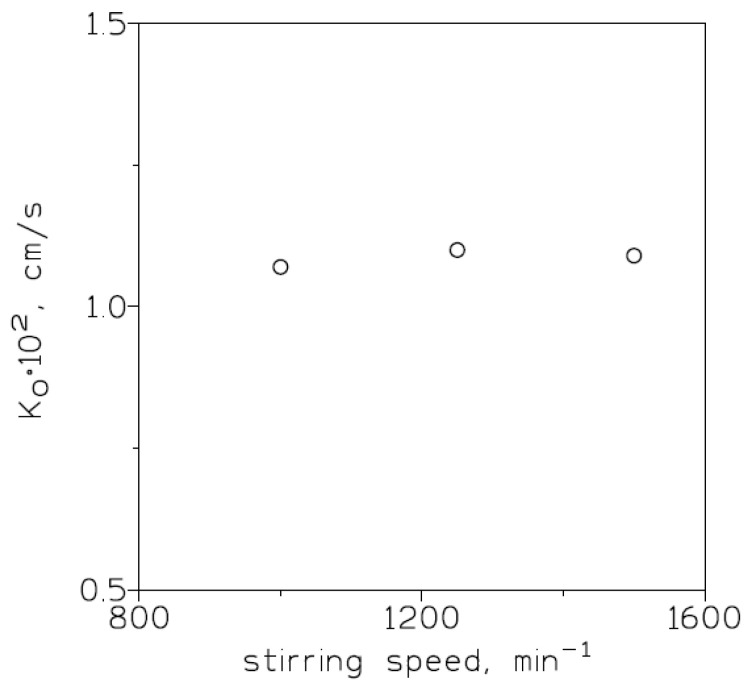
Overall mass transfer coefficients versus stirring speeds (feed phase) for chromium(VI) transport using Cyanex 921 in Solvesso 100 as mobile carrier. Support: GVHP4700.

**Figure 3 membranes-13-00177-f003:**
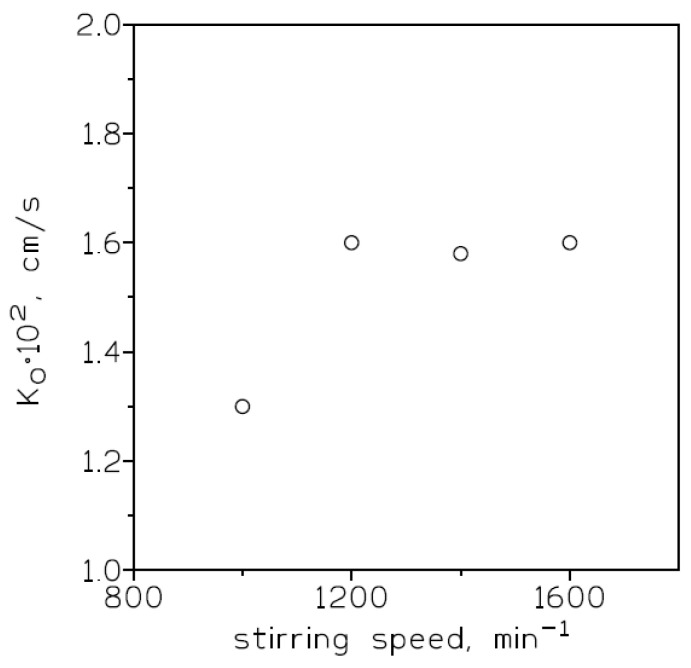
Overall mass transfer coefficients versus stirring speeds (feed phase) for chromium(VI) transport using Cyanex 923 in Solvesso 100 as mobile carrier. Support: GVHP4700.

**Figure 4 membranes-13-00177-f004:**
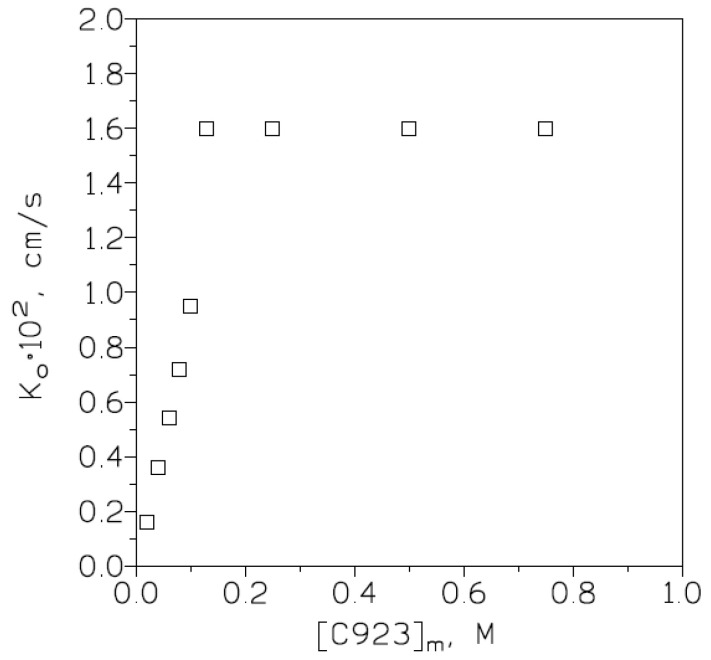
Variation in K_o_ versus carrier concentration in the membrane phase. Membrane support: GHVP4700. Temperature: 20 °C.

**Figure 5 membranes-13-00177-f005:**
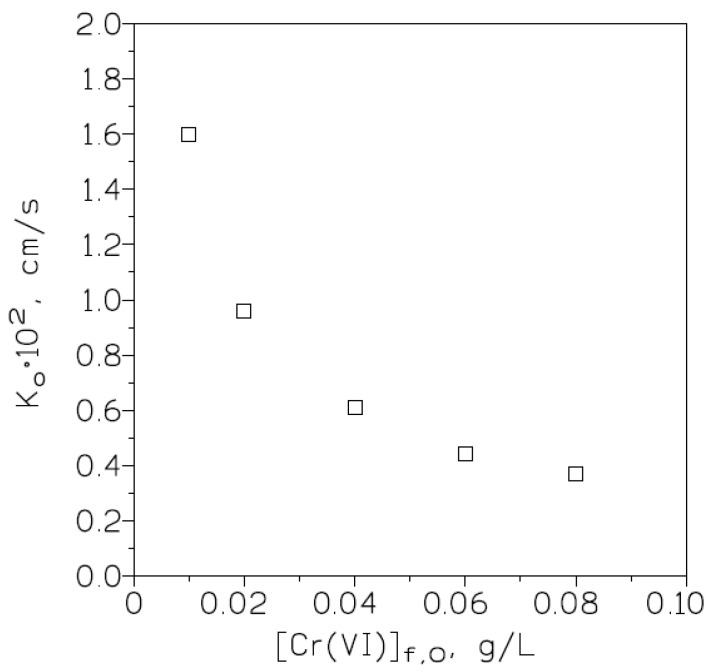
Plot of K_0_ versus initial chromium(VI) concentration in the feed phase for metal transport investigations using a 0.5 M Cyanex 923 in Solvesso 100 solution as carrier phase.

**Table 1 membranes-13-00177-t001:** Influence of HCl concentration on chromium(VI) transport.

HCl, M	K_o_·10^2^ cm/s
0.25	0.83
0.5	0.99
0.75	1.1
1	1.1
2	1.1

Feed phase: 0.01 g/L Cr(VI) in HCl medium. Temperature: 20 °C.

**Table 2 membranes-13-00177-t002:** Transport of Cr(VI) at various Cyanex 921 concentrations in the membrane phase.

[Cyanex 921], M	K_o_ · 10^2^, cm/s
0.01	0.14
0.02	0.28
0.04	0.54
0.06	0.79
0.13	0.99
0.25	1.0
0.38	1.0
0.5	1.1

Membrane phase: Cyanex 921 in Solvesso 100 immobilized on GVHP4700 support. Temperature: 20 °C.

**Table 3 membranes-13-00177-t003:** Chromium(VI) transport at various initial metal concentrations.

[Cr(VI)]_f,0_ g/L	K_o_·10^2^, cm/s
0.01	1.1
0.015	0.97
0.025	0.88
0.05	0.85
0.075	0.79
0.1	0.52

Feed phase: Cr(VI) in 1 M HCl. Membrane phase: 0.5 M Cyanex 921 in Solvesso 100 immobilized on GVHP4700 support. Stripping phase: 5 g/L hydrazine sulphate. Temperature: 20 °C.

**Table 4 membranes-13-00177-t004:** Values of log K_ext_ and n for chromium(VI) extraction by Cyanex 921 and Cyanex 923.

	Cyanex 921	Cyanex 923
log K_ext_ (graphical)	3.34	1.94
n (graphical)	1.94	1.94
r^2^	0.9530	0.9763
log K_ext_ (numerical)	3.48 ± 0.21	3.27 ± 0.14
n (numerical)	2	2
U	0.132	0.058

**Table 5 membranes-13-00177-t005:** Diffusional parameters of the Cr(VI)-Cyanex 921 and Cr(VI)-Cyanex 923^−^transport systems.

Parameter	Cyanex 921	Cyanex 923
Mass transfer resistance due to membrane phase (s/cm)	175	396
Mass transfer resistance due to feed phase (s/cm)	111	114
Diffusion coefficient in the membrane phase (cm^2^/s)	7.1 × 10^−5^	3.3 × 10^−5^
Diffusion coefficient in the bulk of the membrane phase (cm^2^/s)	2.6 × 10^−4^	1.3 × 10^−4^
Metal flux (mol/cm^2^s) ^a^	1.9 × 10^−9^	3.0 × 10^−9^
Apparent diffusion coefficient in the membrane phase (cm^2^/s) ^b^	9.5 × 10^−8^	1.5 × 10^−7^
Limiting flux (mol/cm^2^ s) ^c^	7.2 × 10^−7^	3.2 × 10^−7^
Mass transfer coefficient in the feed phase (cm/s)	9.0 × 10^−3^	8.8 × 10^−3^

Values for maximum chromium(VI) transport at 1 M HCl in the feed phase. ^a^ Calculated with the overall mass transfer coefficients obtained when 0.25 M carrier concentrations and an initial metal concentration in the feed phase of 0.01 g/L are used. ^b^ Calculated using 0.25 M carrier concentrations in the membrane phase and metal flux values shown in the table. ^c^ Calculated at 0.25 M carrier concentrations in the membrane phase.

## Data Availability

Not applicable.

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
