# Peer review of "Transport of Chromium(VI) across a Supported Liquid Membrane Containing Cyanex 921 or Cyanex 923 Dissolved in Solvesso 100 as Carrier Phase: Estimation of Diffusional Parameters"

_membranes, 2023, doi:10.3390/membranes13020177_

Round 1
Reviewer 1 Report
Dear Editor-in-Chief:
Membranes
23- Jan- 2023
----------------------------------------------------------------------------------------------
Manuscript ID: membranes-2194195
"Transport of chromium (VI) across a supported liquid membrane containing Cyanex 921 or Cyanex 923 dissolved in Solvesso 100 as carrier"
Thank you so much for choosing me to review this manuscript, Kindly, find the attached comments and my recommendation. Please feel free to contact me if you require further information.
Recommendation: minor revision
Comments to author(s):
This manuscript presents an important study on the transport of chromium (VI) across a supported liquid membrane containing Cyanex 921 or Cyanex 923 dissolved in Solvesso 100 as carrier. In general, the manuscript is interesting but there are many basic issues within the paper to address in detail to improve the article's quality:
Introduction
a) In the introduction, you can referee for other methods for chromium removal and other heavy metals as you can cite:
(2023) Studies on the potential use of activated carbon from guava seeds (AC-GS) as a prospective sorbent for the removal of Cr(VI) from aqueous acidic medium, International Journal of Environmental Analytical Chemistry, 103:2, 378-395, DOI: 10.1080/03067319.2020.1858071.
1. Masry, B. A., Elhady, M. A., & Mousaa, I. M. (2022). Fabrication of a novel polyvinylpyrrolidone/abietic acid hydrogel by gamma irradiation for the recovery of Zn, Co, Mn and Ni from aqueous acidic solution. Inorganic and Nano-Metal Chemistry, DOI: 10.1080/24701556.2022.2034860.
b) You should include some references for chromium removal by emulsion liquid membrane
and other metal ions such indium, palladium and silver, you can cite:
1. Masry, B. A., Aly, M. I., & Daoud, J. A. (2021). Selective permeation of Ag+ ions from pyrosulfite solution through Nano-Emulsion Liquid Membrane (NELM) containing CYANEX 925 as carrier. Colloids and Surfaces A: Physicochemical and Engineering Aspects, 610, 125713.
2. Kassem, A. T., Masry, B. A., Zeid, M. M., Noweir, H. G., Saad, E. A., & Daoud, J. A. (2017). Extraction of palladium from nitrate medium by emulsion liquid membrane containing CYANEX 471X as carrier. Solvent Extraction and Ion Exchange, 35(2), 145-160.
- Results and discussion
- The author should study the effect of membrane porosity on the permeation of chromium and indium
- The author should include comparison between their concluded results using Millipore Durapore GVHP4700 (polyvinylidene fluoride) supported membrane with other types of membranes such as cellulose nitrate and mixed cellulose ester.
In view of the above, I can recommend the manuscript to publish in the membranes journal after addressing the items mentioned above.
Best Regards
Sincerely yours
Dr. Botros. A. Masry, Ph.D.
Chemistry of Nuclear fuel department
Hot Laboratories and Waste Management Center,
Atomic Energy Authority,
Post Office 13759, Cairo, Egypt
E-mail: betaam2@yahoo.com,
Botros.masry@eaea.org.eg

Reviewer 2 Report
The article "Transport of Chromium(VI) across a Supported Liquid Membrane Containing Cyanex 921 or Cyanex 923 Dissolved in Solvesso 100 as Carrier Phase: Estimation of Diffusional Parameters" corresponds to the subject of the journal and can be published, but only after revision.
The introduction requires an almost complete rewrite. It contains very few references to previous research. Much is incomprehensible. In addition, it is necessary to clearly formulate the purpose of the study and justify its relevance.
More detailed comments on the introduction are provided below.
1. 1. Page 1, lines 28-38. Here, perhaps, it is worth introducing more specifics, which will allow a reader who is unfamiliar with the described problem to better understand it. So, it is possible to describe what exactly is the harm to a person from the ingestion of chromium (VI) into his body. It would be nice to give numerical values for MPC in wastewater, concentrations that are considered dangerous for humans. All this will make it possible to more reasonably formulate the relevance of this study.
2. 2. Page 1, lines 39-43. Please provide links to relevant studies on the extraction of chromium (VI) by various methods.
3. 3. Page 1, lines 39-43. In this paragraph, it is worth talking about which methods are not only being researched, but are actively used in production. What methods have found industrial application and have already proven their effectiveness?
4. 4. Page 2, lines 44-45. Here it is necessary to talk about the advantages of liquid membranes over other methods. Why is the use of liquid membranes better than extraction, adsorption, etc.?
5. 5. Page 2, lines 47-51. Please provide links to the types of liquid membranes listed.
6. 6. Page 2, lines 54-59. It lacks a description of previous research in this area. Why do you use Cyanex 921 and Cyanex 923 for impregnation? Have they already been studied in the literature? What other liquid membranes have been used to extract chromium (VI) before you? How is this work different from others (doi.org/10.1016/j.chemosphere.2004.07.019, doi.org/10.1016/S0304-386X(01)00147-5, doi.org/10.1016/j.memsci.2009.03.049, doi.org/10.1002/jctb.903 etc.)?
7. Page 2, lines 54-59. Why is the essence of the work reduced to the study of experimental variables? Why is it necessary to study diffusion parameters? Please discuss the importance of your work. This will require references to similar similar studies.
Smaller remarks relate to the main part of the article:
8. Page 3, figure 1. The word "membrane" obviously contains a misprint. The inscriptions containing the formulas of chromium and hydrochloric acid should be moved so that they do not overlap the vertical lines.
9. If possible, please present the results of the experiments described in sections 3.1.1 - 3.1.3, 3.2.1 - 3.2.4 in the form of graphs.
10. Please indicate how long the experiments were carried out? To what extent is the system under study stable over time? Are impregnating liquids washed out?
